# PIBNet: a Physics-Inspired Boundary Network for Multiple Scattering Simulations

## Abstract

The boundary element method (BEM) provides an efficient numerical framework for solving multiple scattering problems in unbounded homogeneous domains, since it reduces the discretization to the domain boundaries, thereby condensing the computational complexity. The procedure first consists in determining the solution trace on the boundaries of the domain by solving a boundary integral equation, after which the volumetric solution can be recovered at low computational cost with a boundary integral representation. As the first step of the BEM represents the main computational bottleneck, we introduce PIBNet, a learning-based approach designed to approximate the solution trace. The method leverages a physics-inspired graph-based strategy to model obstacles and their long-range interactions efficiently. Then, we introduce a novel multiscale graph neural network architecture for simulating the multiple scattering. To train and evaluate our network, we present a benchmark consisting of several datasets of different types of multiple scattering problems. The results indicate that our approach not only surpasses existing state-of-the-art learning-based methods on the considered tasks but also exhibits superior generalization to settings with an increased number of obstacles. Code available upon acceptance.

## 1 Introduction

Multiple scattering is defined by Martin (2006) as "the interaction of fields with two or more obstacles", as illustrated in fig. 1. The main challenge in such problems lies in the combined effect of the number of obstacles and their separation distances, as these factors govern the complexity of the multiple reflections. The boundary element method (BEM) (Bonnet, 1999) is an efficient numerical method for solving linear partial differential equations (PDEs) such as the one involved in multiple scattering. Unlike the finite element method (FEM), the BEM reformulates the PDE into a boundary integral equation (BIE) with unknowns restricted to the problem boundaries. After solving the BIE, the solution in the volumetric domain is evaluated using the boundary integral representation. Thus, by reducing the dimensionality of the problem, the BEM can offer significant gains in computational efficiency and accuracy compared to the FEM for wave propagation problems in unbounded domains. Within the BEM, the two computational stages have very different costs: solving the BIE on the boundaries is significantly more expensive than evaluating the boundary integral representation to recover the volumetric solution.

To reduce the overall computational cost, several learning-based approaches have been proposed to simulate multiple scattering (Hao et al., 2021; Nair et al., 2025). Taking advantage of the recent development in neural network architectures for solving PDEs, these approaches rely either on discretizing the solution domain or on neural fields. Discretization-based methods (Pfaff et al., 2020; Zhdanov et al., 2025) handle complex geometries and boundary conditions but inherit the computational cost of traditional solvers. Neural field approaches (Raissi et al., 2019) represent the solution as a continuous function, allowing inference at arbitrary points, but for complex geometries, they still rely on conditioning over discretized domains (Serrano et al., 2024; Alkin et al., 2024; 2025). Due to these limitations, learning-based multiple-scattering methods have been restrained to two-dimensional problems only.

Inspired by methods based on the BEM that learn the boundary solution instead of solving the BIE to alleviate the BEM bottleneck (Lin et al., 2021; Fang et al., 2024), we propose, PIBNet. It is a

learning-based method to simulate PDEs that can be solved with the BEM i.e. linear and semi-linear PDEs. More specifically, in this paper, we address multiple scattering due the inherent challenges of such problems. We first present a benchmark including simulation datasets that focus on 3D exterior scattering problems for both Helmholtz and Laplace problems under Dirichlet or Neumann boundary conditions, as well as meaningful metrics to evaluate the estimated solution. Neural fields, which have frequently been used to solve similar tasks (Lin et al., 2021; Fang et al., 2024), are not suitable here due to the presence of multiple disjoint obstacles. Since efficiently capturing long-range interactions between obstacles is crucial, a natural approach is to explore transformer-based methods developed for solving PDEs or point cloud processing. However, as shown numerically in section 6.2, these approaches exhibit limited performance. To improve the accuracy, we propose PIBNet, a method in which distant interactions are explicitly represented as edges in graphs. To avoid the computational cost of adding all possible interaction edges, we propose a physics-informed strategy for connecting distant points, combined with a multiscale message-passing graph neural network (GNN) that approximate the boundary solution to multiple scattering problems. In contrast to other message-passing architectures for PDE simulation, such as Lino et al. (2022), the work most closely related to ours, we strictly apply the principles of U-Net (Ronneberger et al., 2015), thereby increasing the latent dimensionality at coarser resolutions. To this end, we introduce a node feature expander that increases the latent dimensionality at coarser resolutions. This approach outperforms existing state-of-the-art learning-based methods dedicated to solving PDEs. Finally, we show the ability of our architecture to generalize to environments with up to three times more obstacles than those seen in the training set. In summary, we propose the following contributions:

1. A benchmark of datasets of exterior Laplace and Helmholtz problems with multiple disjoint obstacles obtained using the BEM.

2. PIBNet, our method that combines a physics-inspired strategy for modeling long-range interactions as graphs and a new U-Net-like multiscale GNN architecture for estimating the boundary solution of the multiple scattering problems.

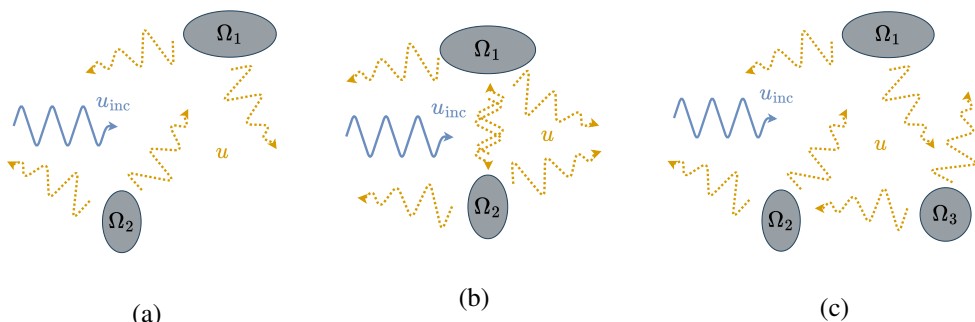

(a)          (b)         (c)

Figure 1: Illustrations of the resulting field $u$ (dashed arrows) from the scattering of an incoming wave $u_{inc}$ (solid arrows) by two obstacles $\Omega_1$ and $\Omega_2$ (a). In (b), the obstacles are closer to each other and in (c), there is a third obstacle $\Omega_3$. This induces more complex interactions between $u_{inc}$ and obstacles.

## 2    PRELIMINARIES ON THE BOUNDARY ELEMENT METHOD

This section provides a comprehensive overview of the boundary element method (BEM) (Bonnet, 1999) through the problem of multiple scattering (Martin, 2006) of an incident wave $u_{inc}$. Let $\Omega = \bigcup_{i=1}^{n} \Omega_i \in \mathbb{R}^3$ be the union of $n$ closed bounded sets $\Omega_i$, $1 \leq i \leq n$, representing obstacles that do not intersect. We assume that each set $\Omega_i$ has a Lipschitz-continuous and piecewise-smooth boundary $\Gamma_i$ and let $\Gamma = \bigcup_{i=1}^{n} \Gamma_i$. The homogeneous Helmholtz equation is given by:

$$\begin{cases} \mathcal{L}u = 0 & \text{in } \mathbb{R}^3 \setminus \Omega \\ u = -u_{inc} & \text{on } \Gamma \end{cases} \tag{1}$$

where $\mathcal{L} = \Delta + k^2$ with $k$ the wavenumber. We assume the Sommerfeld radiation condition: $\lim_{|\mathbf{x}| \to \infty} |\mathbf{x}| \left( \frac{\partial}{\partial |\mathbf{x}|} - \mathrm{i}k \right) u(\mathbf{x}) = 0$ is satisfied, which ensures that no energy is radiated from infinity. Therefore, the total field is given by $u_{\text{tot}} = u + u_{\text{inc}}$.

The BEM relies on a reformulation of eq. (1) as a boundary integral equation. The key ingredient in this reformulation is the Green's function $G$, defined as the solution of $\mathcal{L}G = \delta$ where $\delta$ denotes the Dirac delta function. For our problem, the variational form of the boundary integral equation can be defined as follows:

$$\int_{\Gamma} u(\mathbf{x})q(\mathbf{x})d\mathbf{x} = \int_{\Gamma \times \Gamma} G(\mathbf{x} - \mathbf{y})q(\mathbf{x})p(\mathbf{y})d\mathbf{x}d\mathbf{y} \tag{2}$$

where $q$ is a test function and $p$ is the unknown. The solution of this equation gives only the trace of the density $p$. The second step of the method consists in applying the boundary integral representation to compute the scattered field in the volume:

$$u(\mathbf{x}) = \mathcal{S}(p)(\mathbf{x}), \quad \mathbf{x} \in \mathbb{R}^3 \setminus \Omega \tag{3}$$

where $\mathcal{S}(p)(\mathbf{x}) = \int_{\Gamma} G(\mathbf{x} - \mathbf{y})p(\mathbf{y})d\mathbf{y}$ is the single layer potential operator.

After discretization of $\Gamma$, eq. (2) becomes a fully populated linear system, with storage and solution complexities of $O(N^2)$ and $O(N^3)$, respectively. Instead of direct solvers, iterative solvers such as GMRES (Saad & Schultz, 1986) are often employed for BIEs. The number of iterations needed for convergence provides insight into the intrinsic difficulty of the problem, such as the level of multiple scattering occurring between obstacles in the given configurations. Fast algorithms like the Fast Multipole Method or Hierarchical Matrices (Chaillat et al., 2008; 2017; Darve, 2000) bring the complexity of BEM down to linear or quasi-linear, yet the method remains computationally expensive for parametric investigations involving many configurations or frequency ranges.

The computational bottleneck of the BEM lies in solving the boundary integral equation. Once the solution trace on the boundary is obtained, the volumetric field can be efficiently reconstructed using the boundary integral representation. We develop a machine learning model to predict the boundary solution, allowing us to accelerate the most expensive part of the computation while still recovering the full volumetric solution.

## 3 RELATED WORK

### 3.1 LEARNING AND BOUNDARY ELEMENT METHOD

Many approaches aim to replace the resolution of the BIE with a neural network, since it is the most computationally demanding part of the BEM. Most of these approaches are conditional neural fields (Lin et al., 2021; 2023; Qu et al., 2024). In these models, the inputs consist of 2D coordinates of points on the surface, together with additional information specific to the problem, such as surface shape parameters or boundary conditions. To improve accuracy, Fang et al. (2024) and Meng et al. (2024) use a frequency representation for input values with a high variation range. More complex geometries have also been addressed with GNN in Wang et al. (2025b). Once the solution of the BIE is returned by the neural network, it can be used to compute the solution of the original PDE with the boundary integral representation as in BEMs. Thus, learning-based BEM can provide an efficient framework for solving PDEs since the solution domain dimension of the BIE is smaller than that of the original PDE. However, all the approaches discussed in this section are limited to problems involving a single surface. This work addresses scattering problems with multiple disjoint obstacles, a setting that introduces interesting challenges.

### 3.2 LEARNING PDEs ON UNSTRUCTURED DATA

PDEs often require processing unstructured data, either because of complex domains (airfoils, geological formations) or to leverage adaptive meshing (Pfaff et al., 2020). Computer-vision-based approaches that are effective with grid-structured data (Wang et al., 2025a; Colagrande et al., 2025) can be adapted to unstructured data with continuous convolutions (Ummenhofer et al., 2019), for

instance. Other works (Li et al., 2023a;b) project the unstructured data onto a regular grid in the latent space before applying FNO layers (Li et al., 2021). As mentioned in the previous section, neural fields may be leveraged (Fang et al., 2024) but they are limited by the capacity of the conditioning mechanism to address complex geometries. Many GNN approaches have been developed (Sanchez-Gonzalez et al., 2020; Li et al., 2019; 2020a;b). One of the most popular, MeshGraphNet (Pfaff et al., 2020), has been extended several times to handle multiple resolutions (Fortunato et al., 2022; Cao et al., 2023; Ripken et al., 2023). However, none of these approaches expand latent dimensions for low-resolution meshes. Following U-Net (Ronneberger et al., 2015), the multi-level extension of MeshGraphNet we propose in this paper implements this feature. More recently, multiple transformer-based methods reach top performance on numerous benchmarks, most of which focusing on reducing the quadratic computation complexity of attention. Thus, several approaches propose efficient attention mechanisms (Li et al., 2022; Hao et al., 2023; Xiao et al., 2024). Other approaches propose to apply transformers locally. For instance, the Transolver architecture (Li et al., 2022; Luo et al., 2025) performs attention on learnable slices of flexible shape of the input data. Erwin (Zhdanov et al., 2025) and GOAT (Wen et al., 2025) clustered the input data in hierarchical balls. Finally, as shown in (Zhdanov et al., 2025), attention-based point cloud processing approaches (Wu et al., 2024b) may be relevant for learning PDEs with unstructured data. In this paper, based on our knowledge of the BEM, we propose a neural architecture to approximate the boundary solution of multiple scattering problems that outperforms other learning-based approaches designed for solving PDEs on unstructured data.

## 4 DATASETS & METRICS

This section presents the benchmark employed for solving the following three problems:

1. Exterior Helmholtz Dirichlet problem with an incident wave of unit amplitude emitted from a monopole source. The Dirichlet boundary condition is therefore parametrized by the source location $\mathbf{x}_0 \in \mathbb{R}^3 \setminus \Omega \cup \Gamma$ and the wavenumber $k$:

$$u(\mathbf{x}) = -\frac{\mathrm{e}^{\mathrm{i}k\|\mathbf{x}-\mathbf{x}_0\|_2}}{\|\mathbf{x}-\mathbf{x}_0\|_2}, \quad \mathbf{x} \in \Gamma. \tag{4}$$

2. Exterior Helmholtz Neumann problem with an incident unit plane wave. The Neumann boundary condition is parametrized by the incident wave's direction $\mathbf{v}$ and the wavenumber $k$:

$$\frac{\partial u}{\partial \mathbf{n}} = -\mathrm{i}k\mathrm{e}^{\mathrm{i}k\mathbf{x}\cdot\mathbf{v}}, \quad \mathbf{x} \in \Gamma \tag{5}$$

where $\frac{\partial u}{\partial \mathbf{n}} = \nabla u(\mathbf{x}) \cdot \mathbf{n}$ is the normal derivative and $\mathbf{n}$ the normal vector to $\Gamma$ at $\mathbf{x}$.

3. Exterior Laplace Dirichlet problem with standard boundary conditions:

$$u(\mathbf{x}) = -\Phi_0 - \frac{\Phi_1}{\|\mathbf{x}-\mathbf{x}_0\|_2} - 2\Phi_2\mathbf{v} \cdot \frac{\mathbf{x}-\mathbf{x}_0}{\|\mathbf{x}-\mathbf{x}_0\|_2}, \quad \mathbf{x} \in \Gamma \tag{6}$$

where $\Phi_0$, $\Phi_1$ and $\Phi_2$ are three constants between $-1$ and $1$, $\mathbf{x}_0 \in \mathbb{R}^3 \setminus \Omega \cup \Gamma$ and $\mathbf{v}$ is a unit vector in $\mathbb{R}^3$.

We generated a training dataset and several test datasets for each problem. Each sample consists of randomly sized and positioned, non-overlapping ellipsoidal obstacles, randomly selected boundary condition parameters, and the corresponding boundary solution trace as the label. Ellipsoids are characterized by their center coordinates and semi-axis lengths.

For each problem, the training set samples contain three obstacles, and we created a test set with three obstacles. Moreover, for Laplace and Helmholtz Dirichlet problems, we also provide test sets with six and nine obstacles. The meshes representing the obstacles and the trace solution were generated using the BEMPP library (Betcke & Scroggs, 2021). We also recorded the number of GMRES iterations required to converge for each sample to monitor their computational complexity. Further details about the datasets are provided in table 1, and the BIE formulations used to generate the labels are included in the appendix (see appendix A).

For evaluation, we assess the performance on the Laplace problem using the relative error of the trace, $\mathrm{Err}_{\mathrm{rel}}$. For the Helmholtz problems, we introduce two metrics: the relative error of the trace

Table 1: Main characteristics of our datasets. Lengths are given without unit.

| Number of samples in the training / test sets | Environment size | Edges length in obstacle meshes | Ellipses semi-axes length (min - max) | Wavelength (min - max) |
|---|---|---|---|---|
| $10^4$ / $10^3$ | $10 \times 10 \times 10$ | 0.1 | 0.3 - 1.5 | 0.6 - 6 |

amplitude, $\text{Err}_{\text{ampl}}$, and the absolute error of the trace phase, $\text{Err}_{\text{angle}}$. The definitions of these metrics for a single sample are given by:

$$\text{Err}_{\text{rel}} = \sum_{x \in \Gamma} \left| \hat{p}(x) - p^*(x) \right| / \left( \sum_{x \in \Gamma} |p^*(x)| \right) \tag{7}$$

$$\text{Err}_{\text{ampl}} = \frac{1}{\#\Gamma} \sum_{x \in \Gamma} \left| \frac{|\hat{p}(x)| - |p^*(x)|}{|p^*(x)|} \right| \tag{8}$$

$$\text{Err}_{\text{angle}} = \frac{1}{\#\Gamma} \sum_{x \in \Gamma} \text{atan2}(\sin(\Delta p), \cos(\Delta p)), \quad \Delta p = \angle \hat{p}(x) - \angle p^*(x) \tag{9}$$

where $p^*$ and $\hat{p}$ denote the ground-truth boundary trace obtained from the BEM and the neural network prediction, respectively, $\#$ indicates the cardinality of a set, and $\angle$ stands for the angle of a complex number. These metrics are then averaged over all samples in a dataset.

## 5 METHOD

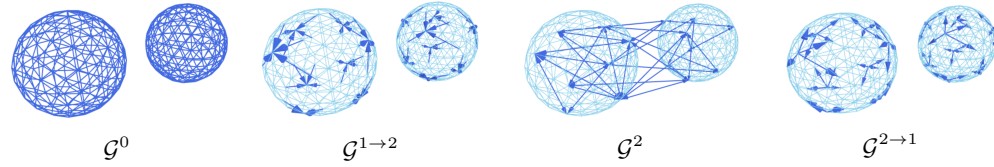

$$\mathcal{G}^0 \qquad\qquad \mathcal{G}^{1 \rightarrow 2} \qquad\qquad \mathcal{G}^2 \qquad\qquad \mathcal{G}^{2 \rightarrow 1}$$

Figure 2: Illustration of some directed graph representations used in PIBNet in the case of two obstacles. Graph edges are in dark blue, while the light blue corresponds to the obstacle meshes. $\mathcal{G}^0$: the *boundary graph* (arrows have been omitted for readability), $\mathcal{G}^{1 \rightarrow 2}$: the *downsampling graph* from level 1 to 2, $\mathcal{G}^2$: the *distant nodes graph* and $\mathcal{G}^{2 \rightarrow 1}$: the *upsampling graph* from level 2 to 1. We omit the downsampling and upsampling graphs $\mathcal{G}^{0 \rightarrow 1}$ and $\mathcal{G}^{1 \rightarrow 0}$ between levels 0 and 1 which are similar to graphs $\mathcal{G}^{1 \rightarrow 2}$ and $\mathcal{G}^{2 \rightarrow 1}$ at a lower resolution.

This paper focuses on the task of multiple scattering Martin (2006), therefore, an accurate representation of the obstacles and of their precise relative position is mandatory. For this purpose, we build graphs representing the obstacles at various resolution levels and graphs for the transitions between two levels. At the highest level (i.e. the lowest resolution), we create a graph connecting distant obstacles with a physics-inspired edge selection strategy. In this section, we present our PIBNet method: after defining the different graphs and the physics-inspired edge selection strategy, we introduce our GNN architecture which consists of an encoder and a processor. The encoder initializes the graph nodes and edges features, while the processor apply message-passing layers to model the interactions between the obstacles.

**Graphs definition** Let $\mathcal{G}^0 = (V^0, E^0)$ be the graph called *boundary graph* with nodes $V^0$ and bidirected edges $E^0$ that corresponds to the union of the $n$ meshes $\mathcal{G}^0_{\Gamma_i} = (V^0_{\Gamma_i}, E^0_{\Gamma_i})$ that represents the discretized boundaries $\Gamma_i$ of the obstacles $\Omega_i$, $1 \leq i \leq n$, so that $\mathcal{G}^0 = \bigcup_{i=1}^n \mathcal{G}^0_{\Gamma_i} = (\bigcup_{i=1}^n V^0_{\Gamma_i}, \bigcup_{i=1}^n E^0_{\Gamma_i})$. We build a multiscale representation of $\mathcal{G}^0$ with $L$ levels using an octree so that for each level $0 < j < L$, we have a new set of nodes $V^j$ satisfying $p(V^j) \subset p(V^{j-1})$ where $p(V)$ returns the set of positions of the nodes in the node set $V$. We also obtain downsampling directed edges $E^{j-1 \rightarrow j}$ and upsampling directed edges $E^{j \rightarrow j-1}$ that link nodes between level $j - 1$ and $j$. The downsampling edges are used to create *downsampling graphs*

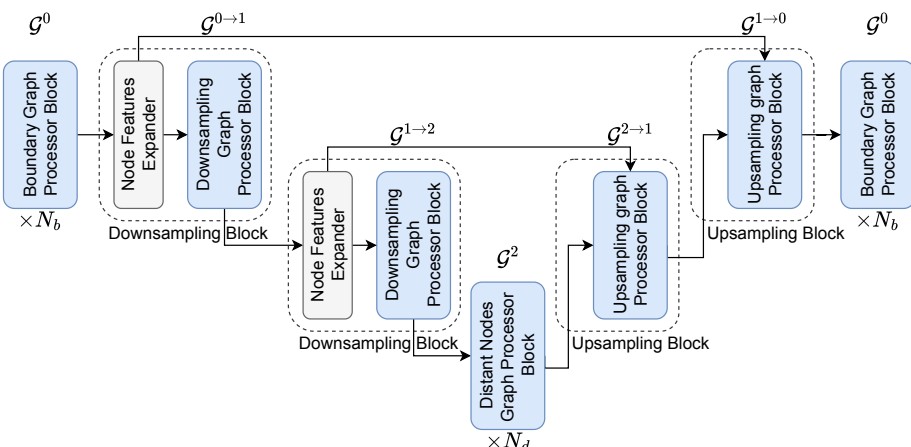

Figure 3: Illustration of the processor part of PIBNet GNN architecture with a number of levels $L = 3$. The graph representation used by each processor block is given above the corresponding block. It is composed of $N_b$ Boundary Graph Processor Blocks at the beginning, then 2 Downsampling Blocks, $N_d$ Distant Nodes Graph Processor Blocks, $L - 1$ Upsampling Blocks, and $N_b$ Boundary Graph Processor Blocks at the end.

$\mathcal{G}^{j-1 \rightarrow j} = (V^{\mathrm{d}, j-1}, E^{j-1 \rightarrow j})$ where $p(V^{\mathrm{d}, j}) = p(V^j)$ while the upsampling edges provides *upsampling graphs* $\mathcal{G}^{j \rightarrow j-1} = (V^{\mathrm{u}, j-1}, E^{j \rightarrow j-1})$ where $p(V^{\mathrm{u}, j}) = p(V^j)$. To model the interactions between distant nodes, we create a *distant nodes graph* $\mathcal{G}^{L-1} = (V^{L-1}, E^{L-1})$ with bidirected edges $E^{L-1}$ so that each nodes of the highest level $V^{L-1}$ is connected to a fraction $\alpha$ of the other nodes in $V^{L-1}$ according to a physics-inspired strategy described below. We illustrated some of these graphs in fig. 2.

**Physics-inspired edge selection strategy** In this paper, we address the Laplace and Dirichlet problems, for which the Green functions are given by $G(\mathbf{x}) = -\frac{1}{4\pi r}$ and $G(\mathbf{x}) = -\frac{e^{-ikr}}{4\pi r}$, respectively, where $r = \|\mathbf{x}\|_2$, $\mathbf{x} \in \mathbb{R}^3 \setminus \Omega \cup \Gamma$. This implies that obstacles interact more strongly at short distances, while long-range interactions remain present. Thus, to generate the edges $E^{L-1}$ in the *distant nodes graph* $\mathcal{G}^{L-1}$, we propose an edge selection strategy consistent with physics that favors connections between close nodes but still allows some longer-range interactions. For each of the $\alpha|V^{L-1}|$ required edges originating from a node in $V^{L-1}$, $n_c$ candidate edges are randomly proposed from the $V^{L-1} - 1$ potential candidates, and the shortest one is chosen.

**Encoder** The encoder initializes the edge features of each graph and the node features of $\mathcal{G}^0$; the node features of the other graphs are initialized later. The edge features of each graph are encoded using a dedicated MLP. Thus, for each edge $e$ of a graph $\mathcal{G}$, the inputs of the encoder are a sinusoidal positional encoding of the distance $D(e)$ between the two connected nodes, the normalized direction of $e$, and other features which depend on the boundary conditions (more details in appendix B). The output dimension is $d^0$ for edges in $E^0$, $d^j$ for edges in $E^{j-1 \rightarrow j}$ and $E^{j-1 \rightarrow j}$, and $d^{L-1}$ for edges in $E^{L-1}$. The node features of $V^0$ are encoded with an MLP whose inputs are determined by the boundary conditions (more details in appendix B). The output dimension of this MLP is $d^0$.

**Processor** The processor, illustrated in fig. 3, is composed of multiple processor blocks that are applied to a graph $\mathcal{G} = (V, E)$. Each processor blocks consists of a message-passing with one edge feature update eq. (10) followed by one node feature update eq. (11):

$$f'_{e_{kl}} = \phi^e(f_{e_{kl}}, f_{v_k}, f_{v_l}) \tag{10}$$

$$f'_{v_k} = \phi^n(f_{v_k}, \sum_k f'_{e_{kl}}) \tag{11}$$

where $\phi^e$ and $\phi^n$ are MLPs and $f_{e_{kl}}$ are the features of edge $e_{kl} \in E$ connecting node $v_k \in V$ to node $v_l \in V$ of features $f_{v_k}$ and $f_{v_l}$, respectively.

¡The first step of the processor consists of $N_b$ processor blocks applied to the boundary graph $\mathcal{G}^0$. It is followed by $L-1$ downsampling blocks, each applied to a downsampling graph $\mathcal{G}^{j-1 \to j}$, $0 < j < L$. To this end, the features of nodes $V^{\mathrm{d},j-1}$ are initialized by the node features expander, which is an MLP that projects the node features of $V^{\mathrm{d},j-1}$ (or of $V^0$ when $j = 0$) from a dimension $d^{j-1}$ to a dimension $d^j$, before a processor block is applied. Then, the node features of $V^{L-1}$ are initialized with the output node features of the final downsampling block and $N_d$ processor blocks are applied to the distant nodes graph $\mathcal{G}^{L-1}$. Thereafter, $L-1$ upsampling blocks are applied, one to each upsampling graph $\mathcal{G}^{j \to j-1}$, that is composed of a processor block that reduces the feature dimension from $d^j$ to $d^{j-1}$. The node features of $V^{\mathrm{u},j-1}$ are initialized with the node features of $V^{\mathrm{u},j}$ (or $V^{L-1}$ if $j = L-1$) for the nodes that exist in both sets and with the node features of $V^{\mathrm{d},j-1}$ otherwise. Finally, $N_b$ processor blocks are applied to the graph $\mathcal{G}^0$ whose node features are initialized with the node features of $V^{\mathrm{u},1}$, output by the last upsampling block. The last processor block returns an output vector of dimension $d^f$ for each node in $V^0$ which corresponds to the approximate BIE solution on the discretization of the boundary $\Gamma$. For the Laplace problem $d^f = 1$ and for the Helmholtz problem $d^f = 2$ corresponding to the real and imaginary parts of the BIE solution, respectively.

## 6 EXPERIMENTS

We evaluate PIBNet through extensive experiments on the new benchmark proposed. We compare the results of our method against those of other state-of-the-art learning-based approaches for solving PDEs on unstructured data including GNN-based methods such as MeshGraphNet (Pfaff et al., 2020), MuS-GNN (Lino et al., 2022), BSMS-GNN (Cao et al., 2023) or transformer-based methods such as Transolver (Wu et al., 2024a), Transolver++ (Luo et al., 2025) and Erwin (Zhdanov et al., 2025). For the fairness of comparison, all GNN-based baselines are trained and evaluated on graphs that incorporate the additional edges introduced by our physics-inspired edge selection strategy to connect the various obstacles. We also include the point cloud processing method Point Transformer v3 referred to as PTv3 (Wu et al., 2024b) in our comparisons.

### 6.1 IMPLEMENTATION DETAILS

Regarding our architecture, we set the number of levels to $L = 3$ and we connect each node to a fraction $\alpha = 0.1$ of the other nodes at the highest level $L-1$. The expansion rate of the dimension of latent layers is 2 when the level index increases by one. This means that for $0 < j < L$, the dimension $d^j$ of a latent layer is given by $d^j = d^0 \times 2^j$. Moreover, we set the dimension of the latent layers of the first level to $d^0 = 64$. We perform the octree using Ripken et al. (2023) implementation and keeping every two levels. In the physics-inspired strategy, the number of candidate edges $n_c$ per required edge in $E^{L-1}$ is set to 2 unless otherwise mentioned. The architecture details of the methods we compare against are given in appendix C. All models are supervised with the groundtruth solution of the BIE using a Huber loss (Huber, 1992) with parameter $\delta = 1.0$. The models are trained from scratch for 100 epochs, with AdamW optimizer (Loshchilov & Hutter, 2017), a batch size of 16, a learning rate starting at $10^{-4}$ decreasing to $10^{-7}$ with a cosine scheduler, and gradient clipping by norm with a maximum value of $1.0$. We also leverage data augmentation which consists of the same random rotation applied to the node positions of the input mesh and to the source location. Hardware details are given in appendix D.

### 6.2 RESULTS

In table 2, we present the performance on the test datasets with the same data distribution as the training set, i.e., with three obstacles per sample. Since our approach involves random processes, we report the mean of five evaluations, each with a different seed. We measure an average relative standard deviation of 0.3% across all datasets and metrics, demonstrating the stability and reproducibility of our method. The results highlight the superiority of GNN approaches over transformer-based methods. This may be due to the better expressiveness of graph representations which explicitly model distant interactions with edges, compared to multihead attention. Furthermore, the results show that our physics-inspired edge selection strategy is architecture agnostic. The lower performance of BSMS-GNN is due to its GNN architecture which relies on graph convolutions providing a coarser modeling of distant interaction relative to message passing with edge feature updates as in the other GNNs. Our PIBNet method outperforms MeshGraphNet Pfaff et al. (2020), the second-best approach,

Table 2: Benchmark results on three obstacle datasets.

| Method | Number of Parameters (M) | Laplace $\text{Err}_{\text{rel}}$ | Helmholtz Dirichlet | | Helmholtz Neumann | |
|---|---|---|---|---|---|---|
| | | | $\text{Err}_{\text{ampl}}$ | $\text{Err}_{\text{angle}}$ | $\text{Err}_{\text{ampl}}$ | $\text{Err}_{\text{angle}}$ |
| MeshGraphNet | 2.4 | **0.039** | 0.116 | 0.114 | 0.053 | 0.052 |
| MuS-GNN | 2.7 | 0.0632 | 0.161 | 0.149 | 0.068 | 0.069 |
| BSMS-GNN | 0.8 | 0.107 | 0.221 | 0.189 | 0.080 | 0.083 |
| PTv3 | 77.2 | 0.066 | 0.148 | 0.141 | 0.065 | 0.068 |
| Transolver | 4.6 | 0.183 | 0.596 | 0.454 | 0.070 | 0.073 |
| Transolver++ | 4.2 | 0.195 | 0.635 | 0.466 | 0.103 | 0.122 |
| Erwin | 7.9 | 0.119 | 0.218 | 0.175 | 0.074 | 0.076 |
| Ours | 5.2 | 0.041 | **0.091** | **0.090** | **0.046** | **0.047** |

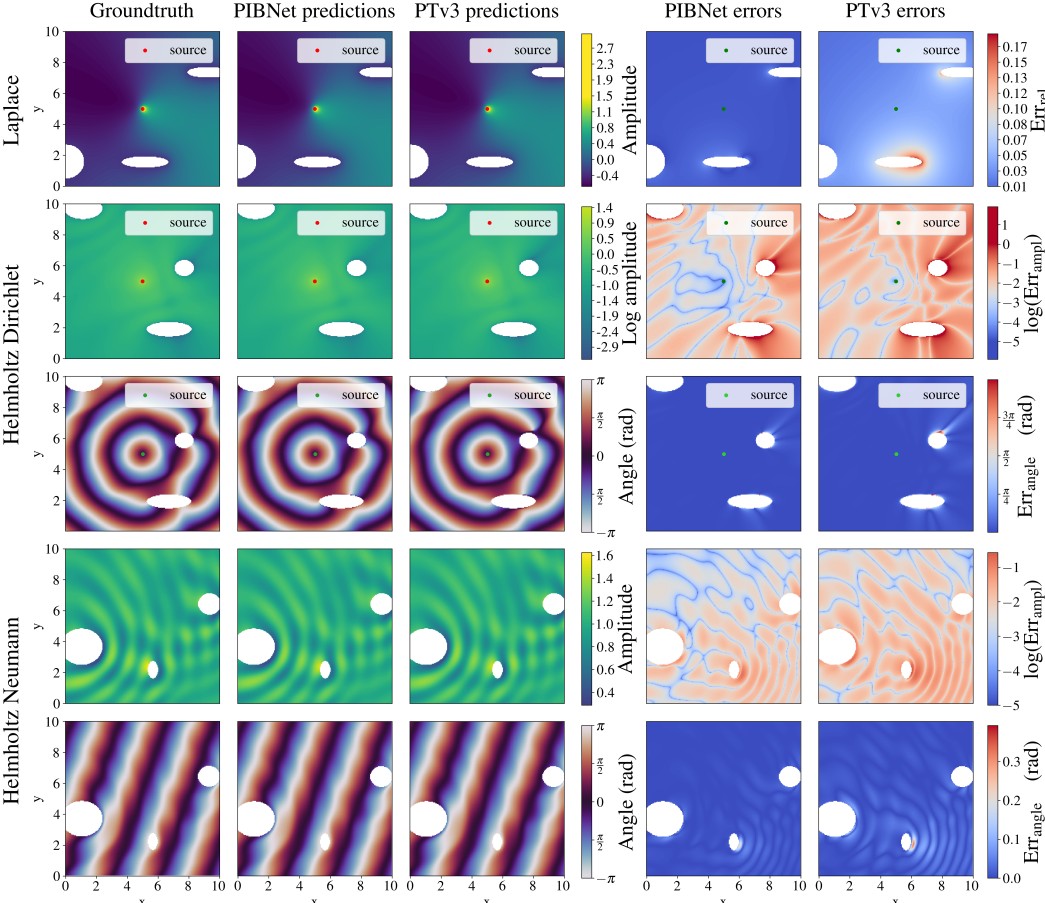

Figure 4: Qualitative results for the three problems of our benchmark with three obstacles (in white). From left to right are the 3D volumetric solution of the total field obtained with GMRES (the groundtruth), with PIBNet and with PTv3 predictions of the trace solution, respectively, and the corresponding errors relative to the groundtruth with PIBNet and PTv3, respectively. For each problem, the 3D volumetric solutions of the total field and their associated errors are sampled within a square domain of side length 10 on the plane $z = 0$.

with an average improvement of 8% across all datasets and metrics. The fact that MeshGraphNet outperforms MuS-GNN may be explained by the latter's hierarchical architecture in which the latent dimensionality is held fixed across resolutions, thereby limiting network capacity at coarser levels. In contrast, our architecture increases the latent dimension at low resolutions, leading to the best performance. PTv3 is the best transformer-based approach; its good performance may stem from its larger number of parameters, but it could also be attributed to its architecture, which is specifically

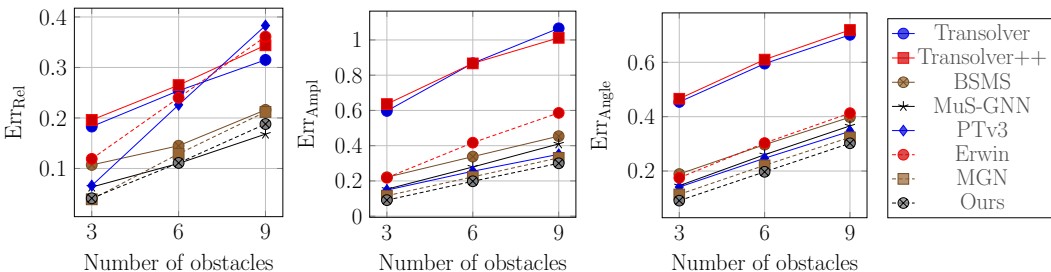

Figure 5: Estimation errors as a function of the number of obstacles for the Laplace Dirichlet problem (left) and the Helmholtz Dirichlet problem (middle and right).

designed for processing point clouds. The irregular distribution of points in point cloud datasets aligns more closely with our data, which consists of points sampled on the surfaces of distant objects. In contrast, PDE benchmarks (Pfaff et al., 2020; Janny et al., 2023) are typically based on nearly regular meshes that cover almost the entire domain. We also note that Transolver (Wu et al., 2024a) and Transolver++ (Luo et al., 2025) are not adapted for this task since they reach lower performance than our baseline in table 3 which do not model distant interaction between obstacles. In appendix E, we analyze the correlation between the per-sample mean absolute error and the GMRES iteration count (a proxy for the number of reflections), the wavenumber and the dispersion of obstacles. Qualitative results for the top GNN and transformer approaches that are PIBNet and PTv3, respectively, are shown in fig. 4.

## 6.3 GENERALIZATION

In fig. 5, we examine the generalization ability of models trained on datasets with three obstacles per sample to test environments with six and nine obstacles. As the number of obstacles exceeds three, we adapt the physics-inspired edge selection strategy during inference to adjust the focus on the closest obstacles. Specifically, the number $n_c$ of candidate edges per required edge in $E^{L-1}$ is increased to 3. A thorough study of the impact of $n_c$ on performance for different numbers of obstacles is provided in appendix F. As the number of obstacles increases, the results show a consistent decline in performance across all methods. This trend is expected because a higher number of obstacles induces quadratic growth in pairwise interactions. This substantially increases the problem's complexity, resulting in reduced performance in scenarios with six or nine obstacles. Additionally, our models are only trained with a fixed number of obstacles per dataset sample, which may affect their ability to generalize to a different number of obstacles. PIBNet demonstrates that it generalize better than the other methods on wave problems. For the Laplace problems, PIBNet's lower generalization capabilities relative to MuS-GNN can be explained by the overfitting of our model with scenarios with three obstacles. Since our PIBNet architecture has been designed for more complex Helmholtz problems, the PIBNet capacity may be too high for the comparatively simpler Laplace problems.

## 6.4 RUNTIME ANALYSIS

In fig. 6, we study the runtime for generating the trace of the solution on the boundaries for the Helmholtz Dirichlet problem with respect to the number of obstacles. We compare approaches running on the same GPU including learning-based methods and the BEM with different convergence tolerance thresholds rtol for GMRES. This study shows that learning-based approaches are orders of magnitude faster than the BEM. However, unlike learning-based methods, whose runtime depends only on the number of nodes in the obstacle meshes, the BEM's runtime is sensitive to wavelength and the relative position of obstacles. Regarding learning-based methods, the very low runtime of Transolver and Transolver++ reflects minimal computational cost, as both approaches are designed to process very large meshes (Wu et al., 2024a; Luo et al., 2025), which explains their limited performance. For BSMS-GNN, the low runtime relative to other GNN-based methods stem from the absence of MLP to process edge features. PIBNet has the highest runtime for three- and six-obstacle problems, but remains close to MuS-GNN, Point Transformer V3 and Erwin. However, we observe that PIBNet scales better than Erwin, which has the worst runtime for nine-obstacle problems. The

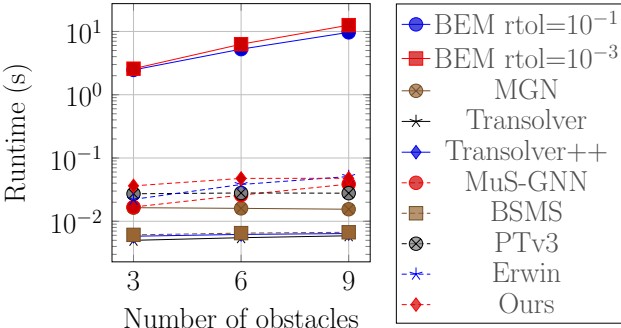

Figure 6: Comparison of runtime on with respect to the number of obstacles for learning-based methods and for the BEM considering different different convergence tolerance thresholds rtol for GMRES.

higher runtime of PIBNet relative to other graph methods is mainly due to the increase of latent dimension at coarser scales.

### 6.5 ABLATION

In table 3, we study the impact of the different components of our method for the exterior Helmholtz Dirichlet problem. First, we highlight the importance of connecting distant nodes to account for dependencies between them. Then, we demonstrate the advantages of using a U-Net-like multi-level architecture with an optimal number of levels, which is achieved with $L = 3$. To be fair in terms of computational complexity, we maintain the same number of edges connecting distant nodes as the number of levels varies. We justify that expanding the dimension of the latent layers at each level results in better performance than keeping it constant even if the average latent dimension is identical. Finally, we show that our physics-inspired edge selection strategy for connecting nodes at the highest level is superior to a purely random edge selection.

Table 3: Ablation study. $n_c$ corresponds to the number of candidate edges per required edge in $E^{L-1}$.

| Distant nodes connection | Number of levels $L$ | Latent dimension at the finest level | Latent dimension expansion factor | $n_c$ | Err$_{\text{ampl}}$ | Err$_{\text{angle}}$ |
|:---:|:---:|:---:|:---:|:---:|:---:|:---:|
| ✗ | 1 | 64 | – | – | 0.328 | 0.205 |
| ✓ | 1 | 64 | – | 1 | 0.202 | 0.181 |
| ✓ | 2 | 64 | 2 | 1 | 0.138 | 0.135 |
| ✓ | 3 | 64 | 2 | 1 | 0.097 | 0.097 |
| ✓ | 4 | 64 | 2 | 1 | 0.115 | 0.112 |
| ✓ | 3 | 128 | 1 | 1 | 0.112 | 0.110 |
| ✓ | 3 | 64 | 2 | 2 | **0.091** | **0.090** |
| ✓ | 3 | 64 | 2 | 3 | 0.093 | 0.092 |

## 7 CONCLUSION

In this work, we introduce PIBNet, a new learning-based method for solving linear and semi-linear PDEs that can be solved by the BEM and focus on multiple scattering problems. This approach proposes to explicitly represent distant interactions that occur in such tasks using graphs created with a physics-inspired strategy. Based on the BEM, we present a new multiscale GNN architecture for estimating the solution trace on the boundary of the domain. Thus, the solution for the entire domain can be quickly estimated using the boundary integral representation of the problem. To evaluate this method, we introduce a novel benchmark of multiple scattering simulations which includes Laplace and Helmholtz problems. Our results demonstrate that PIBNet surpasses other state-of-the-art learning-based methods for solving PDEs and processing point clouds. Additionally, PIBNet generalizes better to settings with a greater number of obstacles on wave problems.

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

# APPENDIX

## A DATASETS DETAILS

The BIE and the representation formulations used for each problem we address are given in table 4. The BIE solution is approximated with GMRES (Saad & Schultz, 1986) with a convergence tolerance of $10^{-5}$.

## B INPUT DETAILS

**Laplace Dirichlet problem**

- Node encoder inputs:
    - A sinusoidal positional encoding of the distance between the current node and $\mathbf{x}_0$,
    - The normalized direction of $\mathbf{x}_0$ relative to the current node,

Table 4: Formulations of the BIE and of the representations used to generate the dataset for each problem where S and N are the single-layer and the hypersingular boundary integral operator, respectively, while $\mathcal{S}$ and $\mathcal{D}$ are the single-layer and the double-layer potential operator, respectively.

| Problem | BIE | Representation |
|---|---|---|
| Laplace Dirichlet | $Sp = u$ | $u = \mathcal{S}p$ |
| Helmholtz Dirichlet | $Sp = u$ | $u = \mathcal{S}p$ |
| Helmholtz Neumann | $Np = \frac{\partial u}{\partial \mathbf{n}}$ | $u = \mathcal{D}p$ |

- The normalized direction $\mathbf{v}$,
- Each term of the boundary condition computed at the current node position.

- Edge encoder inputs:

  - A sinusoidal positional encoding of the length of the edge,
  - The normalized direction of the edge,

**Helmholtz Dirichlet problem**

- Node encoder inputs:

  - A sinusoidal positional encoding of the distance between the current node and the source $\mathbf{x}_0$,
  - The normalized direction of the source $\mathbf{x}_0$ relative to the current node,
  - The wavenumber of the sample,
  - The sine and the cosine of the angle of an incident wave coming from $\mathbf{x}_0$.

- Edge encoder inputs:

  - A sinusoidal positional encoding of the length of the edge,
  - The normalized direction of the edge,
  - The wavenumber of the sample,
  - The sine and the cosine of the angle of an incident wave at the destination node coming from the source node.

**Helmholtz Neumann problem**

- Node encoder inputs:

  - The normalized direction of $\mathbf{v}$ of the incident wave,
  - The wavenumber of the sample,
  - The sine and the cosine of the angle of the incident wave at the location of the node,
  - A sinusoidal positional encoding of the distance between the current node and the average of node positions,
  - The direction of the average of the node positions.

  Here, the use of average node positions is mandatory for methods that are not based on meshes to provide information about node positions relative to each other. For the other problems, this information comes from the distance and direction of the source $\mathbf{x}_0$. For the sake of fairness, this input is used with all methods.

- Edge encoder inputs:

  - A sinusoidal positional encoding of the length of the edge,
  - The normalized direction of the edge,
  - The wavenumber of the sample,
  - The sine and the cosine of the angle of an incident wave at the destination node coming from the source node.

Table 5: Implementation details of the different methods we compare against in this paper. The hyperparameters have been tuned to maximize performance on the Helmholtz Dirichlet problem.

| Model | Parameter | Value |
|---|---|---|
| MeshGraphNet (Pfaff et al., 2020) | Processor depth | 15 |
| | Latent dimension | 128 |
| Point Transformer V3 (Wu et al., 2024b) | Grid Size | 0.2 |
| | Latent dimensions | (64, 128, 256, 512) |
| | Window sizes | (512, 512, 512) |
| | Encoder depths | (2, 2, 2, 6) |
| | Encoder heads | (4, 8, 16, 32) |
| | Encoder patch size | 1024 |
| | Decoder depths | (2, 2, 2) |
| | Decoder heads | (4, 8, 16) |
| | Decoder patch size | 1024 |
| | Stride | 2 |
| Transolver (Wu et al., 2024a) | Latent dimension | 256 |
| | Number of layers | 8 |
| | MPL ratio | 4 |
| Transolver++ (Luo et al., 2025) | Latent dimension | 256 |
| | Number of layers | 8 |
| | MPL ratio | 4 |
| Erwin (Zhdanov et al., 2025) | MPNN dim. | 64 |
| | Latent dimensions | (64, 128, 256) |
| | Window sizes | (512, 512, 512) |
| | Encoder depths | (2, 2, 6) |
| | Encoder heads | (4, 8, 16) |
| | Decoder depths | (2, 2) |
| | Decoder heads | (4, 8) |
| | Stride | 2 |
| | Distance-based attention bias | Enabled |
| BSMS-GNN (Cao et al., 2023) | MPNN dim. | 128 |
| | Scale number | 3 |
| MuS-GNN (Lino et al., 2022) | MPNN dim. | 128 |
| | Scale number | 3 |
| | Number of MPs at each level | (4, 2, 4) |

## C   IMPLEMENTATION DETAILS

The implementation details of the different state-of-the-art methods we compare against are given in table 5.

## D   HARDWARE DETAILS

The data was generated on a 36 Intel Core i9-10980XE (3.00GHz) CPUs with double precision and a tolerance threshold for GMRES set to $\mathrm{rtol} = 10^{-5}$ to obtain a highly accurate groundtruth. The time required to generate our training datasets with 10000 samples depends on the problem and the BIE formulation. Generating the Laplace Dirichlet, Helmholtz Dirichlet and Helmholtz Neumann training datasets took 12, 36 and 96 hours, respectively. All training experiments have been conducted on a single NVIDIA RTX 3090 GPU with 24GB memory. Training times range from 3-4 hours for BSMS-GNN and Point Transformer v3, the later benefiting from flash attention acceleration Dao et al. (2022), to more than 18 hours for Transolver and Transolver++ because of gradient accumulation which is mandatory when the batch size exceeds 1. Otherwise, training time is 8-10 hours for PIBNet, 10-11 hours for MeshGraphNet and 12 hours for Erwin. Note that flash attention cannot be used for training Erwin as its distance-based attention bias position encoding is mandatory to achieve good performance.

# E  ANALYSIS OF THE MODEL PERFORMANCE

## Helmholtz Dirichlet

## Helmholtz Neumann

## Laplace Dirichlet

Figure 7: Mean absolute error of PIBNet predictions per dataset sample as a function of GMRES number of iteration to create the corresponding groundtruth (left) and as a function of the corresponding wavenumber (right) for the Helmholtz Dirichlet problem (top), the Helmholtz Neumann problem (middle) and the Laplace Dirichlet problem (bottom).

In this section, we examine how the performance varies with respect to the number of GMRES iterations, the wavenumber and the obstacle dispersion for each dataset sample.

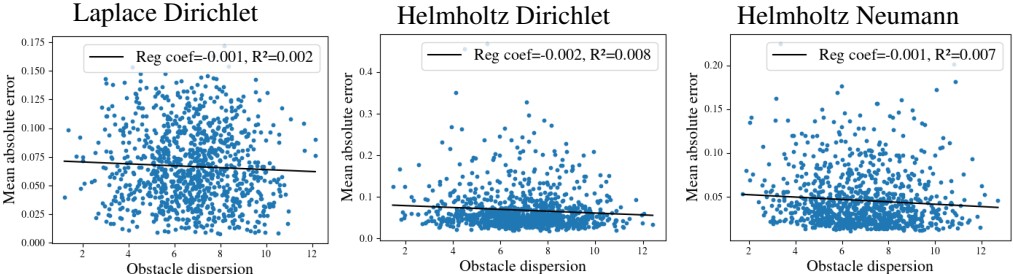

Figure 8: Mean absolute error of PIBNet predictions per dataset sample as a function of obstacle dispersion which is obtained by computing the maximum of the minimum distances between obstacle pairs.

For wave propagation problems, we expect the number of GMRES iterations to reflect the difficulty of the problem, i.e., the number of reflections between obstacles. Consequently, we predict that PIBNet will have more difficulty with this case, which is indeed what we observe in fig. 7 (a, c). In addition, it is well known that the number of iterations tends to increase with the wavenumber. As shown in fig. 7 (right), for Helmholtz problems, both the GMRES iteration count and the PIBNet prediction errors exhibit a similar correlation with the problem's wavenumber. In contrast, PIBNet prediction errors and GMRES number of iterations are weakly correlated for Laplace problems which is not a wave problem.

The fig. 8 shows the PIBNet performance per dataset sample with respect to obstacle dispersion. We quantify dispersion as the maximum of the minimum distances between obstacle pairs for each dataset sample. No correlation is observed for any problem. While this is expected for Laplace problems, for wave problems, closely spaced obstacles are anticipated to produce more complex interactions and thus higher errors. Since most of the error is already explained by the wavenumber, one can compute the partial correlation between prediction errors and obstacle dispersion controlled by the wavenumber. This means that the effect of the wavenumber is removed from the correlation between prediction error and obstacle dispersion. We obtain $-0.09$ and $-0.16$ for the Helmholtz Dirichlet and the Helmholtz Neumann problems, respectively. The two negative partial correlations between error and obstacle dispersion are consistent with what can be expected from wave problems.

## F  IMPACT OF THE NUMBER OF CANDIDATE EDGES AT INFERENCE

Figure 9 illustrates the impact on performance of the number of candidate edges $n_c$ per required edge in graph $\mathcal{G}^{L-1}$ when the number of obstacles varies.

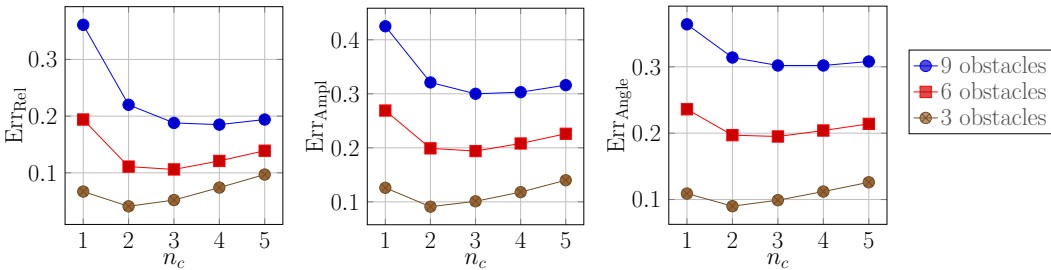

Figure 9: Estimation errors as a function of the number of candidate edges $n_c$ for the Laplace Dirichlet problem (left) and the Helmholtz Dirichlet problem (middle and right).

