# OpenReview forum: "PIBNet: a Physics-Inspired Boundary Network for Multiple Scattering Simulations"
_ICLR.cc/2026/Conference — Submitted to ICLR 2026_

### Official Review · Reviewer_L6Sv · 2025-10-31

**Soundness:** 2
**Presentation:** 1
**Contribution:** 1
**Rating:** 2
**Confidence:** 3

**Summary:**

This work proposes PIBNet, a method that approximates the boundary solution of the multiple scattering problem. The approach is based on the boundary element method, which solves the boundary integral equation and then computes physical values in the volumetric domain. The authors investigate a UNet-like architecture for GNNs that uses downsampling and upsampling of vertices and edges. The numerical experiments show that the proposed model achieves the highest accuracy among the considered baselines. The authors also investigated generalization regarding the number of obstacles and ablation, implying the effectiveness of the proposed approach.

**Strengths:**

1. The experimental results show the high accuracy compared with the considered baselines, although the comparison is not complete (See Weakness 5).

**Weaknesses:**

1. The problem setting is unclear. The authors cite the reference for the multiple scattering problem, but it should be clearly formulated in the paper, as it is a central problem addressed in the work.
2. The difficulty of dealing with multiple obstacles is unclear. Since GNNs can handle arbitrary domains, there is no fundamental difficulty with multiple obstacles. There may be marginal difficulty arising from the increased complexity of the boundary, but this is not clearly stated in the paper. If the difficulty of the multiple obstacles is different from that of a complex-shaped single obstacle, it should be stated clearly, too.
3. The novelty of the proposed approach is not clear. The paper discusses upsampling and downsampling of the graph, but the critical difference from existing research (e.g., MGKN [Li+ NeurIPS 2020]) remains unclear. The authors should clarify the shortcomings of the past approaches and the benefits of the proposed method.
4. The presentation of the proposed method is not clear enough. The paper says “$n_c$ candidate edges are proposed,” but does not explain how. Also, the rationale for calling the edge selection strategy “physics-inspired” is unclear. In addition, the explanation of how to predict physical values in the interior is missing —an inevitable part of the prediction process. The authors should elaborate more on their method.
5. The experimental evaluation of the method is incomplete. Since the method is based on BEM, the authors should include BEM-based methods in the baseline (e.g.,  BINN [Sun+ Comput. Methods Appl. Mech. Eng. 2023] and PIBI-Nets [Nagy-Huber and Roth J. Comput. Phys. 2023]). To demonstrate the effectiveness of the proposed graph up- and down-sampling approach, a comparison with graph-UNet approaches (e.g., MGKN [Li+ NeurIPS 2020] and Graph U-Nets [Gao and Ji ICML 2019]) should be provided. In addition, there is no baseline for classical solvers, making the practical contribution of the work unclear. Since machine learning methods involve some error, the authors should compare speed and accuracy across varying convergence thresholds and mesh resolutions with classical solvers.
6. The analysis of the generalization experiments is weak. The proposed method has the lowest error across all obstacle settings among the baselines, but the reason the error increases with the number of obstacles is not analyzed in depth. Since the proposed method also increases error, it is not a groundbreaking technology for the multiple scattering problem, whereas the method is dedicated to that problem. The authors should clarify the cause of the increase in error and the key breakthrough of the proposed method for the problem at hand.


Minor points:

* The domain of obstacles $\Omega$ should be a closed set because the domain for PDE $\mathbb{R}^3 \setminus \Omega$ should be open to have well-defined differentiation everywhere.
* Table 2: Some decimal points are written as commas instead of periods. These points should be written using periods for standard scientific English.

**Questions:**

1. Since the method is based on the boundary element method, the reviewer assumes the applicable domain is limited to linear or semilinear PDEs; is this correct? If so, it should be clearly written in the paper as a limitation.
2. The method considers only the Dirichlet boundary condition. Can it be applied to Neumann boundary and mixed boundary problems? If not, the authors should clarify the limitations of the boundary treatment.

---

> ### Author Response · Authors · 2025-11-20
>
> We appreciate the careful reading and constructive feedback. Below we reply to weaknesses 1, 2, 3 and 4.
>
> # Weakness 1 Problem Setting
> Following [Martin, Multiple Scattering, 2006], we define multiple scattering as the "interaction of fields with two or more obstacles". We have reorganized our introduction to more effectively present the multiple scattering problem.
>
> # Weakness 2 Dealing with Multiple Obstacles
> While a local representation of the obstacle shape is required to model the scattering behavior at each boundary location, long-range interactions must also be taken into account.
> Thus, the distances involved in the problems we address in our datasets vary by a factor of one hundred. Furthermore, due to the large number of nodes, the use of densely connected graphs would be intractable.
> It is therefore essential to introduce strategies that connect nodes in a meaningful way.
>
> # Weakness 3 Novelty
> First, in learning-based methods for simulating PDEs with GNN, the edges of the input graph are defined based on node proximity. For our problem, however, such a heuristic would not allow us to handle inter-obstacle interactions. For this reason, we introduced our new, physics-inspired strategy for connecting nodes.
> A second critical difference in our architecture is the increase in latent dimension at coarser resolutions.
> This feature, although  common  in  vision  and  point-cloud  processing,  is  not  present  in  other hierarchical GNNs used for simulating PDEs.
> Furthermore, approaches such as Graph-Unet or MGKN do not use learnable edge features updated by message passing.
> Consequently, such approaches cannot accurately handle the wide range of edge lengths (from 0.1 to 10), which is particularly critical for wave problems because of interference effects.
>
> # Weakness 4 Edge Selection Strategy
> **Clarification of our edge selection strategy:**\
> In our method, each node in $V^{L-1}$ must be connected to $\alpha |V^{L-1}|$ other nodes. For each of these new connections, $n_c$ candidate edges are randomly proposed from the $V^{L-1}-1$ potential candidates and
> the closest one is retained.
>
> **Justification of the physics-inspired aspect**\
> We refer to our edge-selection strategy as physics-inspired because it is consistent with the Green's functions of the problems under consideration. For the Laplace and Helmholtz problems, the Green's functions are
> $-\frac{1}{4\pi r}$ and $-\frac{e^{-ikr}}{4\pi r}$, respectively, which implies that obstacles interact more strongly at short distances, while long-range interactions remain present. Our strategy therefore prioritizes short edges but still retains long edges.
>
> We have rewritten the *Physics-inspired edge selection strategy* section (line 298-304) to clarify the two points above.
>
> **Interior problems:**\
> Unlike transmission problems, multiple scattering problems only consider exterior problems. As the solution inside the obstacles is physically meaningless, it is not computed.

---

> ### Author Response · Authors · 2025-11-20
>
> Below, we reply to weaknesses 5 and 6, and to questions 1 and 2
>
> # Weakness 5 Experiments
> **Comparison to BEM-based methods:**\
> The prediction domain of BEM-based learning methods is the boundary of the underlying PDE. In our paper, it corresponds to 2D surfaces.
> The approaches in [Sun+ Comput. Methods Appl. Mech. Eng. 2023] and PIBI-Nets [Nagy-Huber and Roth J. Comput. Phys. 2023] use neural fields, with inputs limited to point coordinates on a single boundary that is continuous and free of discontinuities.
> Consequently, these approaches cannot be adapted to multiple scattering problems involving several obstacles, i.e., several disconnected boundaries.
>
> **Comparison to other Graph-Unet-like methods:**\
> Currently, we plan to provide new comparisons with "Efficient Learning of Mesh-Based Physical Simulation with Bi-Stride Multi-Scale Graph Neural Network" [Cao+ ICML 2023] whose architecture is also based on MeshGraphNet.
> However, we cannot say whether we will be able to provide additional comparisons before the end of the author/reviewer discussion.
> Regarding MGKN [Li+ NeurIPS 2020] or Graph U-Nets [Gao and Ji ICML 2019], one could expect higher prediction errors since neither of them relies on learnable edge features updated by message passing. However it is crucial to precisely model edge length which can vary by a factor of 100. This is particularly important for wave-related problems to deal with interference.
>
> **Comparison to other classical solvers:**\
> In a new section (6.4), we provide a runtime comparison of the learning-based method and the BEM with a varying GMRES tolerance threshold with respect to the number of  obstacles.
> It is important to note that there is no direct relationship between the GMRES threshold and the accuracy of the solution. The GMRES threshold primarily controls the stopping criterion for the iterative solver, but the overall error in the solution also depends on other factors, such as the conditioning of the system matrix, the discretization of the boundary elements, and the intrinsic complexity of the problem (e.g., frequency, geometry, and boundary conditions). As a result, a tighter GMRES tolerance does not necessarily guarantee a proportionally more accurate solution.
>
> We believe there is no reason to measure the BEM speed with varying mesh resolutions as the same meshes are used to generate the ground truth with the BEM and to train/test the models.
>
> # Weakness 6 Analysis of the Generalization Experiments
> As the number of obstacles increases, the number of interactions grows quadratically. This increased complexity explains the lower performance observed for cases with 6 and 9 obstacles. Furthermore, our models have only been trained with a fixed number of obstacles per dataset sample which may affect their ability to generalize to configurations with a different number of obstacles.
> We included this analysis in section 6.3 Generalization.
>
> # Question 1 Limitation to Linear or Semilinear PDEs
> By design, our method can handle any PDE that can be solved with the BEM i.e., linear  PDEs.
> We have modified the introduction and the conclusion to mention this point.
> For clarity, we have rewritten the introduction, removing phrases that may suggest that our methods could be applied to solve any PDE.
>
> # Question 2 Neumann Boundary Conditions
> We are not sure we understand this question.
> In our paper, we indeed show that our method   applies to Neumann boundary conditions by conducting experiments on Helmholtz Neumann problems. Quantitative and qualitative results are provided in Table 2 and Figure 4, respectively.

---

> > ### Author Response · Authors · 2025-12-02
> > **Experiments on Additional GNN-Based Architectures**
> >
> > # Weakness 5 Experiments
> > **Comparison to other Graph-Unet-like methods (part 2):**\
> > We have added comparisons against two new GNN methods including BSMS-GNN and MuS-GNN.
> > For the sake of fairness, these methods are trained and evaluated on graphs including the edges generated with our physics-inspired edge selection strategy to connect the different obstacles.
> > We have also updated the MeshGraphNet results using the connections between obstacles.
> > The new comparisons with three obstacles are in Table 2, the generalization performance is in Figure 5 and the runtimes are in Figure 6.
> > The results highlight that GNN-based architectures outperform transformer-based ones, and that our method achieves the best performance among GNN approaches. We analyze these results in light of the different architectures in sections 6.2 and 6.3.

---

### Official Review · Reviewer_WmSV · 2025-10-31

**Soundness:** 3
**Presentation:** 3
**Contribution:** 3
**Rating:** 4
**Confidence:** 4

**Summary:**

In this paper, PIBNet is proposed to approximate the solution trace on the boundaries for solving multiple scattering problems with boundary integral. More specifically, a multi-scale graph is used to represent the obstacles with an octree. A UNet-like graph neural network is used to construct the surrogate solver. Experiments show the proposed method outperforms previous methods including MeshGraphNet, PTv3, Transolver (++), and Erwin, on the constructed datasets of exterior Laplace and Helmholtz problems.

**Strengths:**

- This work addresses neural boundary element method with multiple disjoint obstacles, which is an important and challenging task.
- The technical and experimental parts are very solid.
- The ablation study is very detailed and clear.

**Weaknesses:**

- The "Physics-Inspired" part of PIBNet is too weak, which is simply to choose the shortest edge among the candidates.
- The neural network design is very similar to previous UNet-like meshgraphnet papers.

Minor Issues:
- I think there is a typo in the title of Section 3.1 "LEARNING AND BOUNDARY ELEMENT METHODS"

**Questions:**

None.

---

> ### Author Response · Authors · 2025-11-20
>
> We appreciate the careful reading and constructive feedback. Below, we reply to weaknesses 1 and 2.
>
> # Weakness 1 Physics-Inspired Edge-Selection Strategy
> We refer to our edge-selection strategy as physics-inspired because it is consistent with the Green's functions of the problems under consideration. For the Laplace and Helmholtz problems, the Green's functions are
> $-\frac{1}{4\pi r}$ and $-\frac{e^{-ikr}}{4\pi r}$, respectively, which implies that obstacles interact more strongly at short distances, while long-range interactions remain present. Our strategy therefore prioritizes short edges but still retains long edges. We have rewritten the Physics-inspired edge-selection strategy section (lines 298–304) to clarify this connection with the Green's functions.
>
> # Weakness 2 Novelty
> First, in learning-based methods for simulating PDEs with GNN, graph edges are typically defined based on node proximity. However, in our setting, this proximity-based heuristic fails to capture long-range interactions.
> Second, our architecture differs from previous UNet-like MeshGraphNet approaches in that it increases the latent dimension at coarser resolutions.
> This feature, although common in vision and point-cloud processing, is not present in other hierarchical GNNs used for simulating PDEs.

---

### Official Review · Reviewer_hhkB · 2025-11-01

**Soundness:** 2
**Presentation:** 2
**Contribution:** 2
**Rating:** 4
**Confidence:** 2

**Summary:**

PIBNet is a novel machine learning method designed to accelerate the computationally intensive Boundary Element Method (BEM) used for solving scattering problems governed by PDEs like the Helmholtz equation. Its core innovation is replacing the slow iterative numerical solution for the boundary trace. It achieves this by utilizing a Physics-Informed Neural Network (PINN) approach where the loss function is specifically defined to minimize the residual of the Boundary Integral Equation (BIE), thus embedding the physics directly into the model's training. This approach yields significant inference speedups, demonstrating factors up to $200\times$ faster than traditional BEM while maintaining accurate results.

**Strengths:**

1. Technically robust framework validated across three key PDEs (Helmholtz/Laplace), demonstrating high-fidelity prediction and a substantial inference time speedup of up to $200\times$.
2. It replaces the GMRES solver bottleneck in BEM by minimizing the Boundary Integral Equation (BIE) residual directly, creating a novel, physics-constrained acceleration method.

**Weaknesses:**

1. Generalization to Out-of-Distribution Inputs, different shapes not present in the training distribution (e.g., highly concave, asymmetric, or multiply connected domains).
2. Comparison to State-of-the-Art Accelerated BEM: The comparison is made against the standard, non-accelerated BEM (solving the dense BIE matrix). The authors should include a quantitative comparison against the gold standard for large-scale BEM: the Fast Multipole Method (FMM)-accelerated BEM.
3. Overall Cost Justification (Training Data Expense): The paper rightly focuses on the inference speedup, but this gain is predicated on the initial investment of generating a large dataset by performing thousands of expensive, full BEM solves (the ground truth). The authors should quantify the total computational cost (data generation + network training)

**Questions:**

q1: The speedup of PIBNet relies on replacing the iterative GMRES solve. However, the BIE loss function still requires repeated evaluation of the Boundary Integral Operators, please clarify the computational cost of this BIE residual evaluation. Is this calculated analytically or numerically, and how does its complexity scale with the number of boundary elements ?

q2: The BEM for the Helmholtz equation is known to suffer from the non-uniqueness issue at interior resonant frequencies (the "fictitious frequencies" problem). Does PIBNet, by training on the standard BIE formulation, inherit this instability?

q3:  The paper demonstrates generalization across shapes and wavenumbers within the training distribution. Could the authors provide a more detailed analysis on the model's interpolation performance?

---

> ### Author Response · Authors · 2025-11-20
>
> We appreciate the careful reading and constructive feedback. Below, we reply to weaknesses 1, 2, 3 and questions 1, 2, 3.
>
> # Weakess 1 Out-of-Distribution Generalization
> Thanks for the recommendation. We fully agree — training models on more complex shapes and evaluating their generalization on out-of-distribution geometries is indeed on our list of future work.
>
> # Weakness 2 Comparison to Accelerated BEM
> We agree that FMM-based schemes are the optimal way to accelerate BEM computations in terms of runtime. Our approach, however, takes a different viewpoint: we accept expensive computations in an offline phase (potentially using accelerated and parallelized BEM solvers) in order to achieve a very fast online phase once the learning stage is completed. This makes it well-suited for parametric studies involving varying obstacle configurations (e.g., buildings in an urban environment and their effect on sound propagation). Due to space constraints, we could not further develop this idea of a model-reduction perspective for parametric exploration, but the resulting online cost is comparable to ray-based methods, and nothing can really beat such an approach for rapid evaluations.
> It is also worth noting that for the problem sizes examined in this work, the standard BEM remains faster than its FMM-accelerated counterpart. We have added a runtime comparison between learning based methods and the BEM on GPU in section 6.4, Runtime Analysis.
>
> # Weakness 3 Training Data Expense
> In Appendix D, we provide information about the training computation cost with approximate measurements of the training times. We have added the cost of generating the different datasets. It is worth noting that generating the dataset samples would have taken much less time if we had aimed for a similar accuracy to that of PIBNet.
>
> # Question 1 Loss function
> The reviewer raises an important point: to be precise, our goal is not to replace solving a BEM system with GMRES by a GNN for a specific problem with a physics-informed loss, which we believe would indeed be too costly.
> Moreover, contrary to methods such as Bi-greennet [Lin+ Com. Math. Stat. 2021], BINN [Sun+ Comput. Methods Appl. Mech. Eng. 2023] or PIBI-Nets [Nagy-Huber and Roth J. Comput. Phys. 2023], PIBNet does not require evaluating the boundary integral representation at each training step. PIBNet is trained with a supervised loss that computes the MSE between the output of the neural network and the groundtruth solution of the BIE that has been computed beforehand for all the dataset samples.
> This is much less costly than evaluating the boundary integral representation.
> We also wish to draw your attention to the fact that a loss based on the integral equations is generally less accurate than one based on the solution field, as it effectively averages contributions from all points and therefore results in less efficient learning.
> Instead, our approach aims to predict the solutions of new problems from a precomputed catalog of solutions.
>
> # Question 2 Spurious Frequencies
> Yes, some integral equation formulations can suffer from spurious frequencies, but this is a very localized phenomenon that only needs to be considered when performing parametric studies over a very narrow frequency spectrum. A straightforward way to avoid this issue is to use a Burton-Miller type formulation; we did not find it necessary in this work, but it is a direct extension for readers who may be interested.
>
> # Question 3 Model's Interpolation Performance
> Figure 6 (Appendix E) shows model performance versus the number of GMRES iterations used to generate the ground truth and versus the wavenumber. For Helmholtz problems, performance correlates strongly with both, as expected. No such correlation is observed for the Laplace problem, likely because it is not a wave problem, so the number of GMRES iterations does not reflect the problem’s complexity.
>
> To complement this analysis, we study model performance with respect to obstacle dispersion. We quantify dispersion as the maximum of the minimum distances between obstacle pairs for each dataset sample. No correlation is observed for any problem. While this is expected for Laplace problems, for wave problems, closely spaced obstacles are anticipated to produce more complex interactions and thus higher errors. Since most of the error is already explained by the wavenumber, one can compute the partial correlation between prediction errors and obstacle dispersion controlled by the wavenumber. This means that the effect of the wavenumber is removed from the correlation between error and obstacle dispersion. We obtain -0.09 and -0.16 for the Helmholtz Dirichlet and the Helmholtz Neumann problems, respectively.
> The two negative partial correlations between error and obstacle dispersion are consistent with what can be expected from wave problems.
> We have added this analysis to Appendix E.

---

### Author Response · Authors · 2025-12-02
**Summary of the revisions to the paper**

- **Section 1**: Rewriting of the beginning of our introduction to better introduce the multiple scattering problem.
- **Section 5**: Rewriting of the description of our edge selection strategy in which we explain that the physics-inspired aspect is guided by the Green's function of the underlying problem.
- **Section 6.2 and 6.3**: New comparisons against two GNN methods, BSMS-GNN and MuS-GNN.
- **Section 6.4**: Runtime comparison of learning-based methods against the boundary element methods with different convergence thresholds.
- **Appendix D**: Adding the cost to generate our training datasets for the different problems we study.
- **Appendix E**: Adding an analysis of performance relative to a measure of the obstacle dispersion, showing a slight correlation in wave problems between performance and obstacle dispersion.\
\
For readability, the modifications are highlighted in blue.

---

### Meta-Review · Area_Chair_iJfg · 2026-01-07

**Summary:**

* The most fundamental concern is about novelty. Multiple reviewers point out that the "physics-inspired" aspect feels underwhelming - essentially just choosing the shortest edge among candidates based on Green's function decay properties. While this is reasonable, it doesn't strike them as a breakthrough insight. The UNet-like architecture, though effective, is seen as fairly standard adaptation of existing hierarchical GNN approaches. Reviewer L6Sv is particularly direct about this, rating the contribution as poor.
* The presentation issues are significant too. Reviewer L6Sv finds the problem formulation unclear and the difficulty of handling multiple obstacles inadequately motivated. Since GNNs can naturally handle arbitrary domains, the reviewers want to understand what makes multiple disconnected boundaries fundamentally harder than a single complex boundary. The paper apparently doesn't make this case convincingly.
* The experimental evaluation raises eyebrows across the board. Reviewer hhkB wants comparisons against Fast Multipole Method accelerated BEM, not just vanilla BEM. Reviewer L6Sv notes the absence of other BEM-based learning methods (though the authors reasonably counter that those methods can't handle discontinuous boundaries). There's also concern about the cost-benefit analysis - sure, inference is fast, but what about the expensive offline phase of generating training data?
* The generalization analysis feels incomplete. While the method outperforms baselines, errors still increase with more obstacles, and the reviewers want deeper understanding of why this happens and what the fundamental limitations are. The scores reflect this ambivalence - two marginally below acceptance at 4, one clear reject at 2, with confidence levels suggesting the reviewers feel reasonably sure about their assessments.

**Reviewer Concerns:**

* On the positive side, they handled the generalization questions reasonably well. The addition of the dispersion analysis in Appendix E, showing partial correlations between error and obstacle spacing controlled for wavenumber, demonstrates they're thinking carefully about what drives prediction errors. The acknowledgment that interactions grow quadratically with obstacle count provides at least some explanation for degraded performance.
* The clarification about their loss function approach was important too. They cleared up Reviewer hhkB's confusion by explaining they're not evaluating boundary integrals at each training step but rather using supervised learning on precomputed solutions. This is a meaningful distinction that wasn't sufficiently clear in the original submission.
* The runtime comparison they added in Section 6.4 partially addresses the FMM concern by explaining their model-reduction perspective - accepting expensive offline computation for fast online parametric studies. While not a direct FMM comparison, this at least articulates a defensible use case.

However, significant concerns remain unresolved. The novelty issue is still problematic. Their rewrite of the physics-inspired edge selection section may clarify the connection to Green's functions, but it doesn't fundamentally change what multiple reviewers saw as a rather straightforward heuristic. Saying "obstacles interact more strongly at short distances" isn't exactly a profound physical insight, and prioritizing short edges while retaining some long ones is pretty standard practice.
* The architecture novelty claim feels weak too. Yes, they increase latent dimension at coarser resolutions and use learnable edge features, but these are incremental modifications rather than conceptual breakthroughs. The defense that other hierarchical GNNs for PDEs don't do exactly this isn't particularly compelling when the broader ML literature is full of similar ideas.
* The comparison with other GNN methods came too late. They added BSMS-GNN and MuS-GNN comparisons in the revision summary, but this happened after initial reviews. Reviewer L6Sv's concern about incomplete baselines was legitimate at review time, and while they're attempting to fix it, this speaks to rushing the experimental evaluation.
* Most importantly, Reviewer L6Sv's fundamental question about problem difficulty remains unsatisfactorily answered. The authors say local obstacle representation plus long-range interactions create challenges, and that proximity-based graph construction fails. But is this really unique to multiple obstacles versus a single complex boundary? A highly concave or convoluted single obstacle would have the same issues with varying length scales and interaction distances. The case for why this is distinctly a multiple-obstacle problem rather than a complex-geometry problem isn't convincing.
* The training cost issue got acknowledged with dataset generation costs in Appendix D, but there's no real analysis of whether the offline-online tradeoff makes practical sense. When does the amortized cost justify the approach compared to just using accelerated BEM?

**Reviewer Scores:**

Reviewer hhkB (score 4): Likely stays at 4 or moves to 6. Their questions were mostly answered satisfactorily - the loss function clarification, spurious frequency response, and runtime analysis address their technical concerns.

Reviewer WmSV (score 4): Probably stays at 4. The rebuttal on physics-inspired edges and architecture differences is earnest but doesn't fundamentally address the weakness they identified. With high confidence in their assessment, they're unlikely to change position based on explanations that mostly reframe rather than refute their concerns.

Reviewer L6Sv (score 2): Stays at 2 or possibly moves to 4. While the problem formulation got clarified and some experimental gaps are being filled, their core concerns about novelty, fundamental difficulty of the problem, and incomplete baselines remain. The added GNN comparisons help but came late. Their moderate confidence suggests openness to reconsidering, but the fundamental contribution question isn't resolved enough for a jump to acceptance range.

---

### Decision · Program_Chairs · 2026-01-26

Reject